# Effects of Non-Thermal Plasma Treatment on Seed Germination and Early Growth of Leguminous Plants—A Review

**DOI:** 10.3390/plants10081616

**Published:** 2021-08-06

**Authors:** Božena Šerá, Vladimír Scholtz, Jana Jirešová, Josef Khun, Jaroslav Julák, Michal Šerý

**Affiliations:** 1Department of Environmental Ecology and Landscape Management, Faculty of Natural Sciences, Comenius University in Bratislava, Ilkovičova 6, 842 15 Bratislava, Slovakia; 2Department of Physics and Measurements, University of Chemistry and Technology Prague, Technická 5, 166 28 Prague, Czech Republic; vladimir.scholtz@vscht.cz (V.S.); jana.jiresova@vscht.cz (J.J.); josef.khun@vscht.cz (J.K.); 3Institute of Immunology and Microbiology, First Faculty of Medicine, Charles University and General University Hospital in Prague, Studničkova 7, 128 00 Prague, Czech Republic; jaroslav.julak@lf1.cuni.cz; 4Faculty of Education, Department of Physics, University of South Bohemia, Jeronýmova 10, 371 15 České Budějovice, Czech Republic; kyklop@pf.jcu.cz

**Keywords:** *Fabaceae*, legumes, low temperature plasma, plasma treatment, seed, seedling

## Abstract

The legumes (*Fabaceae* family) are the second most important agricultural crop, both in terms of harvested area and total production. They are an important source of vegetable proteins and oils for human consumption. Non-thermal plasma (NTP) treatment is a new and effective method in surface microbial inactivation and seed stimulation useable in the agricultural and food industries. This review summarizes current information about characteristics of legume seeds and adult plants after NTP treatment in relation to the seed germination and seedling initial growth, surface microbial decontamination, seed wettability and metabolic activity in different plant growth stages. The information about 19 plant species in relation to the NTP treatment is summarized. Some important plant species as soybean (*Glycine max*), bean (*Phaseolus vulgaris*), mung bean (*Vigna radiata*), black gram (*V. mungo*), pea (*Pisum sativum*), lentil (*Lens culinaris*), peanut (*Arachis hypogaea*), alfalfa (*Medicago sativa*), and chickpea (*Cicer aruetinum*) are discussed. Likevise, some less common plant species i.g. blue lupine (*Lupinus angustifolius*), Egyptian clover (*Trifolium alexandrinum*), fenugreek (*Trigonella foenum-graecum*), and mimosa (*Mimosa pudica, M. caesalpiniafolia*) are mentioned too. Possible promising trends in the use of plasma as a seed pre-packaging technique, a reduction in phytotoxic diseases transmitted by seeds and the effect on reducing dormancy of hard seeds are also pointed out.

## 1. Introduction

Plasma, called also the fourth state of matter, is a partially or fully ionized gas. A distinction is made between thermal (high temperature, equilibrium) and non-thermal (cold, low temperature, non-equilibrium) plasma. The thermal plasma reaches the temperatures of thousands of Kelvins and occurs in the Sun, lighting, electric sparks, tokamaks, etc. and it is therefore not applicable in biological applications. On the other hand, the non-thermal plasma (NTP), also called low-temperature or cold plasma, occurs at nearly ambient temperature and the high kinetic energy is stored in electrons only. Its biological and also medical applications are very wide and include, among others, disinfection processes, acceleration of blood coagulation and improved wound and infection healing, dental applications or cancer therapy. These are summarized in numerous reviews, such as [1,2,3,4,5,6], or in the comprehensive books of Shintani and Sakudo [7] and Metelmann et al. [8].

The use of non-thermal plasmas in agriculture or plant biology has also been widely reported in the last few years. The topics, related to the decontamination of seeds, modification of surface properties, metabolomic pathways, and enzymatic activity, enhancing seed germination and the initial growth, are summarized e.g., in [9,10,11,12,13,14,15,16]. Plant disease control [17] or mycotoxin degradation [18,19] were also reported. The nature of chemical reactions in NTP is rather complex, see e.g., [20,21,22,23].

NTP may be easily generated in various electric discharges, among which the most commonly used ones are corona discharges, plasma jets (called also plasma needles, plasma torches or plasma pens), dielectric barrier discharges, gliding arcs and microwave discharges. For a general description of plasma sources, see e.g., [24,25,26,27]. In addition, the described effects are not constrained to the direct NTP treatment, but on a lower scale are also mediated by the effects of plasma-activated water (PAW) or air, i.e., the water exposed to NTP prior to the application to desired objects. The described effects can persist for many months after exposure, mainly due to the presence of stable reactive oxygen and nitrogen particles as described e.g., in [28,29,30].

The *Fabaceae* family (*Leguminosae*, legumes) are a large group of flowering plants with a worldwide distribution. With some 20,000 species, the *Fabaceae* are the third largest family of higher plants. Members of this family are dominant species in some ecosystems (e.g., *Acacia* sp. in parts of Africa and Australia). They are ecologically important for their ability to symbiotically fix nitrogen [31]. The roots of many *Leguminosae* host distinct and specific symbiotic nitrogen-fixing bacteria (*Rhizobium* sp.). They are sources of oils, timber, gums, dyes, and insecticides [32]. The legumes are second only to cereal crops in agricultural importance based on area harvested and total production [33]. As typical examples, the following species may be mentioned: soybeans (*Glycine max*), peanut (*Arachis hypogaea*), common bean (*Phaseolus vulgaris*), lentil (*Lens culinaris*), pea (*Pisum sativum*); flavouring plants, such as carob (*Ceratonia siliqua*); fodder and soil rotation plants, such as alfalfa (*Medicago sativa*) and clovers (*Trifolium* sp.).

The use of NTP has already been addressed by the authors in relation to wheat (*Triticum aestivum*) [10] and seeds that can be used as raw seed [34]. In this communication, we aim to provide an overview of the rapidly growing amount of knowledge gained by scientific teams in testing the effect of NTP on plant seeds. This manuscript is a continuation to this review trend. We chose seeds of legume plants because they are readily available, are large and therefore easy to work with, and so they can also serve as a model for studying other organisms. In scientific databases, the number of articles dealing with germination and initial growth after plasma application is increasing. The aim of this work was to search, analyze and synthesize contents of scientific articles dealing with the effect of NTP plasma on legumes. Figure 1 summarizes main essentials of our review.

## 2. Surface Seed and Sprout Decontamination

Both microbial and toxin decontamination of seed surface and plant sprout were included. Runtzel et al. [35] reported the effective fungal inactivation of *Aspergillus parasiticus* and *Penicillium* sp. on the surface of common bean after 10–30 min exposure of dielectric barrier discharge (DBD). Selcuk et al. [36] reported the inactivation of pathogenic fungi—*Aspergillus* sp. and *Penicillium* sp.—by NTP in a SF_6_ atmosphere on artificially contaminated seeds of common bean, chickpea, lentil and soybean without affecting the cooking time and other food qualities. A significant reduction of 3-log was achieved within 15 min.

Mitra et al. [37] showed a significant reduction of the initial natural microbial load on the chickpea seed surface of 4.5 ± 0.02 log colony-forming unit (CFU) by 1 and 2-log after 2 and 5 min of NTP treatment. The reduction of *Alternaria* sp., *Mucor* sp., *Fusarium* sp., *Penicillium* sp., *Stemphylium* sp., *Cladosporium* sp. on chickpea and fenugreek was reported in [38], however, without detailed specifications.

A significant reduction of the seed-borne microbial contamination on pea was observed by Khatami and Ahmadinia [39], where the amount of microorganisms decreased by ca 3-log from initial 5.5 to 2.5 log CFU/mL/cm^2^ (the units are not explained in the original paper) after 60 s of exposure. Peanut decontamination was reported by Basaran et al. [40], where 1-log and 5-log reductions of *Aspergillus parasiticus* after 5 min treatment in air or a SF_6_ atmosphere were observed, respectively.

The decontamination of several leguminous species (blue lupine, goat-rue, honey clover, soybean, and pea) has been studied [41,42], where the plasma treatment contributed to better fungicidal effect against of *Fusarium* sp., *Alternaria* sp., and *Stemphilium* sp. on seeds. On the other hand, a possible reduction of toxin production on peanut kernels was reported [43], where a significant reduction of aflatoxin levels without any negative sensory effect was reported.

The decontamination of soybean seeds contaminated with bacteria using PAW was reported by Lee et al. [44]. PAW reduced the overall 4.3-log CFU/mL amount of aerobic microbes and 7.0-log CFU/mL of artificially inoculated *Salmonella* Typhimurium within 5 min and 2 min, respectively. Two following works reported the decontamination of sprouts, both by PAW only. Schnabel et al. [45] contaminated mung beans sprouts with the bacteria *Escherichia coli*, *Pseudomonas fluorescens*, *Pseudomonas marginalis*, *Pectobacterium carotovorum*, and *Listeria innocua*. The experimental results showed a reduction from 2.5-log to 3.5-log of bacteria and better growth of the mung bean sprouts, while untreated samples became strongly glassy and cell liquor was released, no influence of treated samples was observed. Similar results with mung bean were reported also by Xiang et al. [46], where reductions of 2.3 to 2.8-log were observed in aerobic bacteria, yeasts and moulds.

## 3. Effects on Seed Surface Properties

The applications of NTP or PAW also affect the properties of samples. These changes appeared to be beneficial and are listed below, and may be divided into seed surface properties and seed internal content properties. The primary effect on the surface is a decrease of the surface energy leading to better wettability or higher hydrophilicity, as measured by the contact angle of water droplets. This change in the wetting properties of seeds is at least partially due to oxidation of their surface by NTP.

Bormashenko et al. [47,48] reported a contact angle decrease in common beans and their markedly accelerated water absorption after tens of seconds of cold radiofrequency plasma treatment. The treatment leads to hydrophilization of the cotyledon and tissues constituting the seed coat when they are exposed to plasma separately. On the contrary, when the entire seed is exposed to plasma treatment, only the external surface of the common bean is hydrophilized by the cold plasma.

Similar results and a possible explanation were presented by Runtzel et al. [35] who observed on common beans scanning electron microscope (SEM), where both the testa and cotyledon structures showed disruption effects on their cell membranes. The inner surface topography of the cotyledon of chickpea was analyzed by Mitra et al. [37], who observed significant changes in the roughness, leading to a change of membrane permeability. The related conductivity alone increased by more than 100%.

Shapira et al. [49] observed that on lentil seeds this effect is irreversible and that it is not related directly to the electrical charge. Da Silva et al. [50] analyzed the wettability and imbibition of *Mimosa caesalpiniafolia* seeds. The wettability and imbibition were found to be directly related to the treatment duration, probably caused by the chemical alternation of the seeds’ lipid layers. After its complete modification, the increase of wettability saturates, as was also observed after 9 min of exposure. The chemical alternation is probably mainly caused by oxidative processes, as reported on mung beans [51]: the higher effect was obtained in an air atmosphere, while the effect was negligible in He or N_2_.

All these statements are in agreement with the results of [52,53], where the authors observed pea seeds. SEM and Fourier transform infrared (FTIR) surface analyses showed small changes in the surface layer caused by the oxidation of lipids and polysaccharides (the consequences are mentioned in the original work). Moreover, the result of performed genotoxicological tests also confirmed that the level of DNA damage is minimal. A significant increase in water imbibition was also reported for soybean seeds [54,55].

Surface modification was also reported after treatment by PAW. Fan et al. [56] reported a water absorption rate increase from 65% for control samples to 75% for treated mung beans. Sajib et al. [57] reported a similar lipid or wax coat alteration of black gram due to the interactions of NO_2_^−^ and H_2_O_2_ with wax. Zhou et al. [58] confirmed by SEM that the seed coat of mung bean is chapped and that it improves the water and nutrients absorption, which is a condition that could enhance the germination rate of mung bean and promote the growth of hypocotyls and radicles.

## 4. Seed Germination

The NTP treatment can have positive effect on seed germination apparent in an acceleration of germination, an increased germination rate and the breaking of seed dormancy.

Tang et al. [59] found that NTP stimulation significantly increased the germination rate and vigor of alfalfa seeds after 20 s of treatment. Bormashenko et al. [47,48] have found an acceleration of seed germination and germination rate for common beans. Rundzel et al. [35] also found similar effects, improving the germination speed and increasing hypocotyl and radicle length in common beans, after 5 min of treatment exposure. However, after longer exposures (20–30 min) saturation and negative effects occur. Fenugreek was studied by Fadhlalmawla et al. [60], where enhancements of the seed germination rate by 7 and 4 times and growth parameters could occur, probably due to the etching of the seed surface stimulated by the plasma streamers and high electric field. Vejrazka et al. [38] confirmed inhibitory effects on fenugreek, when the highest reduction of germination (over 40%) was recorded after 50 s of NTP treatment.

Mitra et al. [37] also reported increases in the germination percentage, the speed of germination, the shoot and root length, the seedling dry weight, and the vigor index in chickpea, however only for short exposure times. These effects saturate and decrease after 2 min of treatment. Very similar effects were described for mung bean seeds [51,61], pea seeds [39,42,52,62], lentil seeds [47] and peanuts [63], where also a yield improvement by 10% was reported, soybean seeds [54,64] and for many leguminous species, like blue lupine, goat-rue, honey clover, and soybean [41,42]. For red clover seeds, according to Mindaziene et al. [65] this effect is correlated with changes in the phytohormone content, where the amount of abscisic acid decreased and gibberellin/abscisic acid ratio increased.

Moreover, the following two works should be mentioned in more details. Tomekova et al. [66], treated pea seeds with diffuse coplanar surface barrier discharge (DCSBD) plasma operating in air, nitrogen, oxygen and mixtures. Aside from an improvement of germination and growth, DNA damage was also detected. This damage increased with increasing amount of nitrogen, due to the much more intensive UV radiation, and also with increasing treatment time. It was concluded that ambient air seems to be the most suitable atmosphere because of the combination of the plasma chemical composition with water vapour. Svubova et al. [53] declared that the main positive effect on pea seeds, i.e., the overall activation of lytic enzymes in seedlings, is caused by the DCSBD generated in air and nitrogen atmospheres. Increased concentrations of radicals in young seedlings and activation of antioxidant enzymes suggest that low NTP doses act as low stressors, which paradoxically have a stimulating effect on the germination, growth and development of seedlings. Small changes in the surface layer caused by oxidation of lipids and polysaccharides, altering the hydrophilicity and thereby increasing imbibition, were also observed. The DNA damage was minimal after short treatment times. It seems that the positive effect is caused by low doses of NTP stress.

A rapid increase of germination percentage from 5% to 50% after 3 min of exposure was reported for *Mimosa caesalpiniafolia* seeds [50]. This effect may be classified as an overcoming of dormancy due to the effects of NTP on the seed surface. After longer times, this effect decreased. It is generally known that the seeds of the *Fabaceae* family are often dormant. These seeds do not germinate immediately after ripening, but rather they often need an additional stimulus. These seeds germinate well after the disruption of their hard impermeable seed coats. NTP treatments are likely to erode the surface of the seeds so that seeds can absorb water better and begin to germinate faster. Thus, NTP treatment can help dormant seeds with a hard seed coat to break dormancy.

The improvement of seed germination caused by PAW treatment was reported by the following works: for mung beans [56,58,67], where the increase of total phenolic and flavonoid contents was also reported; and for soybean seeds in Lo Potro et al. [68]. The following four works reporting the effects of PAW are mentioned in more details.

In [44], the authors also observed an increase in ascorbate, asparagine and γ-aminobutyric acid (GABA), and followed the development of cotyledon and hypocotyl in germinating soybean seed of soybean from the 1st to the 4th day of cultivation. In Zhou et al. [58], the authors applied PAW on mung beans and reported that the high concentration of ROS might contribute to the chapping of seed coat, which improved its absorption of water and nutrients. The activities of superoxide dismutase (SOD), malondialdehyde (MDA), typical phytohormones affecting the growth of plants indole acetic acid (IAA) and abscisic acid (ABA) contents in mung bean seedlings were followed. It was demonstrated that PAW could reduce membrane lipid peroxidation damage by increasing antioxidant enzyme activities, thus significantly reducing the accumulation of MDA. On the other hand, reactive nitrogen species (RNS, nitrogen oxide NO_X_, HNO_X_) were partially responsible for the acidification of the solution. In [69] authors used PAW and plasma activated liquid fertilizer on lentils. While the PAW caused germination rates as high as 80% against 30% for control ones, the plasma activation of the liquid fertilizer unifies two effects: an early stage boost (probably due to the fertilizer) and an enhancement of growth rate (probably due to the plasma-activated liquid).

Compared to the previous ones, the authors of [70] modified the germination characteristics of pea, common bean, and soybean by plasma coating of the seed surface with macromolecules. To delay germination, two different hydrophobic source gases were utilized: carbon tetrafluoride (CF4) and octadecafluorodecalin (ODFD). Seeds of pea (*P. sativum* cv. Little Marvel and cv. Alaska) treated with CF_4_ displayed a significant delay in germination. Similarly, plasma treatment with ODFD delayed germination in soybean, and common bean seeds. The degree of delay was dependent on the amount of coating applied.

In the recent study of Svubova et al. [71], the effects of cold atmospheric pressure plasma exposure on seed germination of soybean was defined. Seed treatment with lower doses of plasma generated in ambient air and oxygen significantly increased the activity of succinate dehydrogenase (a Krebs cycle enzyme), proving the switching of the germinating seed metabolism from anoxygenic to oxygenic. A positive effect on seed germination was documented, while the seed and seedling vigour were also positively affected.

## 5. Seedling Initial Growth

Affecting of initial seedling growth is closely related to the previously mentioned effects on germination. Thus, the appropriate NTP exposure or the use of PAW causes an initial growth improvement.

The authors of [39] reported an increase of the length of shoots and roots after 30 s and 60 s of plasma treatment with the optimum being observed at 30 s for pea and zucchini (*Cucurbita pepo*, *Cucurbitaceae* family) seedlings. The improvement of root and shoot length, dry weight, and the vigor, together with changes in the production of endogenous hormones (auxins and cytokinins and their catabolites and conjugates) was also reported by Stolárik et al. [52]. In a study by Bußler et al. [62] they determined the effects on the flavonol glycoside profile, considering the impact on their metabolic activity in different growth stages. In 15 day-old seedlings, the concentration of flavonoid glycosides was dose-dependently decreased after two NTP treatments compared to none or three treatments. The photosynthetic efficiency of treated pea sprouts and seedlings declined, indicating a negative effect of NTP treatment on plant metabolism.

For soybean, in [55] the authors report the enhancement of seedling growth and that DBD treatment incremented 1.6 fold the nitrogenase activity in nodules, while leghaemoglobin content was increased two times, indicating that mutualistic bacteria in the nodes fixed nitrogen more actively than the control. Accordingly, the nitrogen content increased by 64% and 23%, respectively, in nodules and the aerial part of plants. In [54], the authors reported a root weight increase by 27% in seed soybean after NTP treatment and that the soluble sugar and protein contents were 16% and 25% higher than those of the control. Soybean seedling growth improvement was reported also in Zhang et al. [72]. Improvements were reported also for black gram [73], where the seed germination rate, seedling growth, total chlorophyll content, total soluble protein and sugar concentrations increased by 13%, 37%, 37%, 53% and 51%, respectively. Similar results were found for red clover [65].

Interesting results concerning the germination and seedling growth under simulated drought stress conditions were reported in the following two works. In [74], the authors found that appropriate NTP alfalfa seed treatment led to increased germination, and the seedlings presented good adaptability to different drought conditions (the consequences are mentioned in the original work). Higher doses had the opposite effect. In Fadhlalmawla et al. [60], the authors showed for fenugreek seeds that plasma treatment affected the growth of seedlings, measured as root and shoot length and fresh and dry weight of root and shoot, but the effect on seedling growth was not consistent. The changes in red clover plant’s internal processes, the beneficial root nodulation and their communication with microorganisms were reported [65,75], where the NTP stress change the amounts of flavonoids important for communication with nitrogen fixing strains of rhizobacteria on the roots of red clover.

Positive effects of PAW were reported in the following works. In Judée et al. [76], the daily irrigation of lentils by PAW led to increases of seedlings length by 34% and 128% after 3 days and 6 days, respectively. In [68], the authors irrigated soybean seeds in soil by PAW. Faster growth and taller soybeans plants were observed, and average stem length values increased from 10 cm to 17 cm. Similar results for black gram were reported in Sajib et al. [57]. For mung beans, the works [56,58] reported positive effects with the optimum of PAW contents, related with the time of water activation and the plasma atmosphere. The optimal activation time was 15 s, as a longer activation led to a decrease of the positive effects. The best positive effects were observed for PAW prepared in air in comparison with that prepared in O_2_, He, and N_2_.

## 6. Seedling Metabolite Activity Affection

The NTP and PAW also affect the properties of the inner contents of seeds or plants related to change in metabolite activity. Bußler et al. [62] studied the effect on pea seedlings’ (*P. sativum* ‘Salamanca’) flavonol glycoside profiles after DBD treatment, while considering the impact on their metabolic activity in different growth stages. Non-acylated and monoacylated triglycerides of quercetin and kaempferol dominated the flavonol glycoside profile, quercetin-3-O-p-coumaroyl-triglucoside being the main flavonoid glycoside. In 15 day-old pea seedlings, the concentration of flavonoid glycosides was dose-dependently decreased in DBD-treated samples. Furthermore, the photosynthetic efficiency of treated pea sprouts and seedlings declined, potentially indicating a negative effect of DBD treatment on plant metabolism.

The oxidative stress caused by NTP and resulting metabolic responses are presented in the following four works. Ebrahimibasabi et al. [77] observed increasing activities of fenugreek catalase by 24%, glutathione peroxidase by 53%, ascorbate peroxidase by 86%. Gebramical et al. [78] reported a decrease in unsaturated fatty acid and moisture content and increased saturated fatty acids, peroxide value, acid value, and total polyphenols in peanuts. Zhang et al. [72] reported significant increases in the activity of the enzymes superoxide dismutase, peroxidase and catalase in soybean sprouts. Stolarik et al. [52] suggested the induction of faster germination and hormonal activities was related to plant signaling and development during the early growth phase of pea seedlings.

The creation of the glycosylation conjugates of high-temperature peanut protein isolate (HPPI) and lactose, improving the solubility of HPPI in peanuts, was reported by Yu et al. [79]. The increase in the degree of glycosylation and a decrease in the degree of browning by DBD accelerated the glycosylation of HPPI and lactose, increasing the solubility and changing the structure of the L-HPPI conjugates. Also, the analysis of protein surface hydrophobicity indicated that the L-HPPI conjugates had a more hydrophilic, stable, and ordered structure (the consequences are mentioned in the original work).

Moreover, Li et al. [80] investigated the impacts of DBD on soybean trypsin inhibitor, which is considered as one of the most important anti-nutritional factors in soybeans. They found that the soybean trypsin inhibitor activities of soymilk were reduced by 86%, probably due to plasma-induced conformational changes and oxidative modification which might contribute to the inactivation of soybean trypsin inhibitor.

Mehr and Koocheki [81] investigated the structure and emulsifying properties of grass pea (*Lathyrus sativus*) protein isolate after DBD treatment. The content of carbonyl groups, dityrosine cross-linking and free sulfhydryl, secondary and tertiary structures, sodium dodecyl sulphate–polyacrylamide gel electrophoresis, surface charge, surface hydrophobicity and solubility of grass pea protein isolate were followed. Overall, the results indicated that cold plasma treatment had positive effects on the interfacial and emulsifying properties of grass pea protein isolate in terms of thermodynamic stability of protein on interface, globulin dissociation, and increase in oil-droplet surface electrical charge.

The goal of the study [82] was to verify the impact of plasma treatment on DNA damage and the induction of positive adaptive responses in pea seedlings. The positive effect of DCSBD (see above) pre-treatment and the reduction of DNA damage of pea were observed at all exposure times used. The strongest repairing effect was observed at exposure times of 120–240 s.

Finally, Lee et al. [44] studied the influence of PAW on soybean cultivation. Its application increases the amount of ascorbate, asparagine, and γ-aminobutyric acid significantly, in the part of cotyledon and hypocotyl of soybean sprouts during 1 to 4 days of farming.

The response of the seeds to NTP together with the use of nanoparticles was tested in the following experiment by Moghanloo et al. [83]. Seeds of *Astragalus fridae* were treated with DBD cold plasma and grown in hormone-free culture medium manipulated with different concentrations of SiO_2_ nanoparticles (nSi). The total dry mass was influenced by different treatments of plasma and nSi, and significant reductions in total dry mass (by 34% and 56%) were observed. The seed treatment with DBD did not cause significant changes in chlorophyll content. Seed treatment with cold plasma and supplementation of rooting medium with nSi led to a severe augmentation in the leaf peroxidase activity when compared to the control. Individual plasma treatments did not produce a significant change in the expression of universal stress protein gene, but the supplementation of culture medium with the different levels of nSi altered the expression rate of this protein. Tissue differentiation patterns (especially vascular system) were affected by the seed treatment with DBD and/or supplementation of rooting medium with nSi.

## 7. Field Production and Quality Crop Yield

The following three works enhance the studies up to the level of field experiments. In Tarrad et al. [84], seeds of Egyptian clover (*T. alexandrium* cv. Gemmiza 1 and cv. Fahl) were treated by pulsed atmospheric-pressure plasma jet, that increased the final yield. The total dry matter yield increased by about 15% and 9% over the non-treated control for Gemmiza 1 and Fahl cultivars, respectively.

In [85], the authors reported for blue lupine NTP treated seed that due to the decrease of seed infection and stimulation of field germination, to early seedling growth and to plant resistance to pathogens, the yield increased by 27%.

In [64], the authors exposed soybean seeds to NTP in various atmospheres of air, O_2_ or N_2_. Under greenhouse conditions, dry weight of roots, plant height, stem diameter and yield of plants grown from either healthy or infected seeds were improved. The plant height, stem diameter and root dry weight of plants from plasma-treated seeds showed increases of 3%, 8% and 12%, respectively; the NTP treatment had positive effects on all the monitored parameters, as compared with either infected plants or fungicide control.

## 8. Miscellaneous Applications

Finally several unclassified but interesting curious works can be mentioned. The first group of works deals with the dry bulk material modulation. In [86], the authors attempted to modify the protein and techno-functional properties of different flour fractions obtained from pea (*P. sativum* cv. Salamanca) seeds. Experiments using a pea protein isolate indicated that the reason for the increase in water and fat binding capacities in protein rich pea flour is based on plasma-induced modifications of the proteins.

Li et al. [80] showed that DBD significantly induced the inactivation of soybean trypsin inhibitor, one of the most important anti-nutritional factors in soybeans, in soymilk and Kunitz-type trypsin inhibitor from soybean in a model system. The soybean trypsin inhibitor activities of soymilk were reduced by 86.1%; the intrinsic fluorescence and surface hydrophobicity of soybean trypsin inhibitor were significantly decreased, while the sulfhydryl contents were increased, so NTP induced conformational changes and oxidative modifications that might contribute to the inactivation of soybean trypsin inhibitor.

Gnapowski et al. [87] also attempted to improve the properties of soybean powder (SBP) suitable as a good food for animals. However, there are two problems with this brew. One is that SBP sinks too fast as parts of SBP are too big and too heavy. Another negative point is a rapid growth of moulds. Their results showed that the NTP is useful to decrease the sinking speed of SBP and no mould growth was observed after the exposure.

The following studies have been devoted to peanuts and their products. Venkataratnam et al. [88] reported the reduction in antigenicity of defatted peanut flour by up to 43% and in whole peanut by up to 9% by the modifications in protein secondary structure caused by NTP. Ji et al. [89,90] exposed peanut protein isolate (PPI) solutions to a DBD plasma. They found a significant improvement in the solubility, emulsion stability, and water holding capacity of PPI, caused by the unfolding of PPI structure, increasing the β-sheet and random coil content and decreasing the α-helix and β-turn content. Moreover, the PPI surface was rougher and more loosely bound, indicating an increase in the PPI specific surface area and exposed protein–water binding sites as well as a marked increase in its oxygen content, suggesting an increase in the hydrophilic groups on the PPI surface. The following study by Ji et al. [91] showed a rapid conjugation between PPI and dextran, caused by changes in the structure of PPI from compact and hydrophobic to loose and hydrophilic.

The last three works present unique and curious topics. In [92], the authors observed that treatment of plant *Mimosa pudica* by NTP induces movements of the pinnules (part of the leaf) and petioles similar to the effects of mechanical stimulation. The gas flow and UV radiation associated with plasma are not the primary reasons for the observed effects, but rather reactive oxygen and nitrogen species (RONS) appeared to be the primary reason for this plasma-induced activation of phytoactuators in plants. Some of these RONS are known to be signaling molecules, which control plants’ developmental processes.

Yepez et al. [93] showed that DBD plasma treatment can transform the liquid soybean oil into a solid product and can produce plasma species that may polymerize polyunsaturated triacylglycerols. Finally, the study of [94] evaluated the influence of PAW on the microbial load and food quality of thin sheets of bean curd (tofu, soybean product). Treatment for 30 min with PAW activated for 90 s reduced the microbial count of total aerobic bacteria and total yeasts and moulds on thin sheets of bean curd.

Table 1 provides an overview of the all monitored plant species and the issues that were studied on them.

## 9. Conclusions

Non-thermal plasma (NTP) has become a widely used technique in various fields of biology, medicine, food processing and others. Different methods are used for its preparation, which, however, makes the comparison of different results somewhat difficult. This review provides a current overview of the effects of NTP on species of the *Fabaceae* family (legumes). The text is divided into logical units, which range from influencing the surface of seeds to affecting whole plants or their products: surface seed and sprout decontamination, seed surface properties affection, seed germination, seedling initial growth, seedling metabolite activity affection, field production and quality crop yield, and miscellaneous applications. This overview shows that NTP may also be well established in various agricultural sectors; here special applications for legumes and their products are covered.

From the works included in this review, the following general conclusions can be drawn: the exposure to NTP or to plasma-activated water (PAW) can significantly affect the different properties of legume seeds. Namely, germination starts from water uptake, and the capability of water absorption could be significantly influenced by the action of plasma. Important surface properties and some physiological parameters of seeds could be also modified. Oxidation processes of plasmatic reactive species may increase water adsorption capability by increasing wettability of seed coats and could also be associated to gas exchanges and to electrolyte leakage by the seed. It is likely that NTP can effectively change dormancy of hard seeds by affecting seed permeability and triggering subsequent processes. NTP can positively influence the germination and growth of the seed, and subsequently also the properties of the seedlings. NTP treatment could reduce the hardness associated with mechanical dormancy of many *Fabaceae* species (alfalfa, blue lupine, grass pea, honey clover, *Mimosa* sp., *Trifolium* sp., etc.). NTP can be advantageously used in decontamination of plant seed surfaces or legume products. Legumes tolerate this physico-chemical treatment well, and the mild stress it causes appears to have a positive effect on them. Changes in physiological factors can then have a positive effect on the number of crops in the field and their yield.

The results discussed in the text are summarized in the Table 1, which provides an overview of previous studies performed with NTP on plant seeds of the *Fabaceae* family.

## Figures and Tables

**Figure 1 plants-10-01616-f001:**
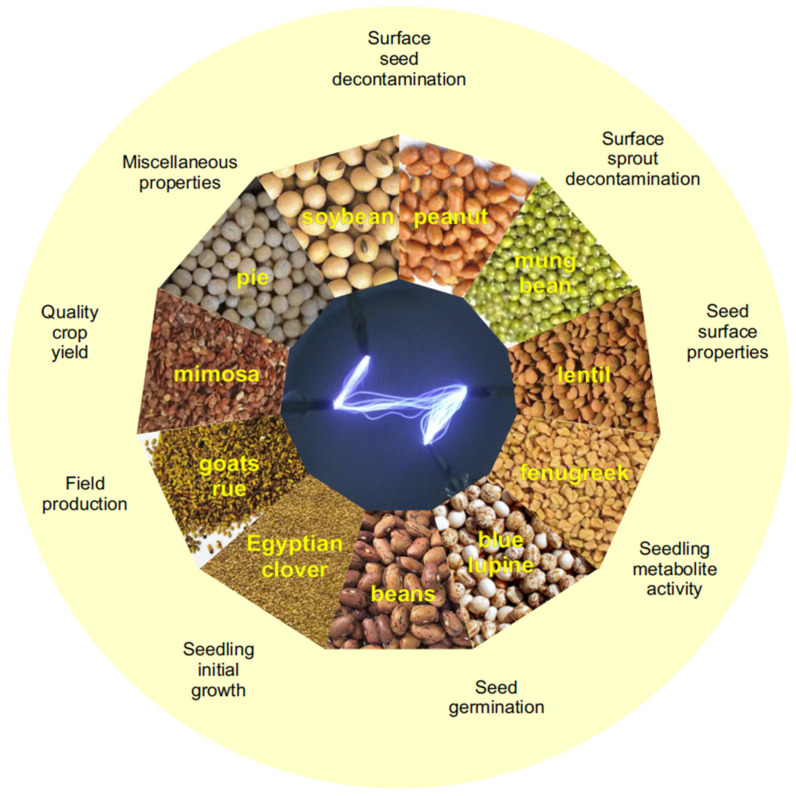
A schematic overview of *Fabaceae* species for which the effect of non-thermal plasma (NTP) has been followed.

**Table 1 plants-10-01616-t001:** Summary of studies performed on species of the *Fabaceae* family or on products thereof after treatment with non-thermal plasma (NTP). The list is sorted alphabetically by plant species, 19 scientific names are mentioned. Abbreviations: syn. synonymous; sn scientific name; DBD dielectric barrier discharge; PAW plasma activated water; AC alternating current; RF radio frequency; USP universal stress protein; SEM scanning electron microscope.

Plant Species	Plasma Source/Device	Object of Study	References
**Alfalfa**, sn: *Medicago sativa* L.	AC glow discharge (20–200 W)	germination	[59]
	Rf glow discharge (air + He mixture, 0–280 W, 15 s)	adaptability of alfalfa seeds in different drought environments	[74]
***Astragalus fridae***, sn: *Astragalus fridae* Rech. F.	DBD (0.84 W/cm^2,^ 0–90 s); applications of SiO_2_ nanoparticle (0–80 mg/l)	expression of USP gene	[83]
**Black gram**, sn: *Vigna mungo* (L.) Hepper	PAW generated by air AC discharge (3–6 kV, 3–10 kHz)	SEM analysis + growth parameter + total soluble protein and sugar concentrations + physiological characteristics + enzyme concentration	[57]
	DBD (0.5 atm, 5 kV, 4.5 kHz, 60 mm, 310 K, 45 W, 20–180 s)	seed cultivar germination + early growth + physiological characteristics	[73]
**Blue lupine** (syn. narrow-leaved lupine), sn: *Lupinus angustifolius* L.	capacitively coupled rf discharge (5.28 MHz, 0.6 W/cm3, 5–20 min)	germination + early growth + surface decontamination	[41,42]
	capacitively coupled rf discharge (5.28 MHz, 0.025 W/cm3, 2–7 min)	microbial reduction + crop yield	[85]
**Chickpea**, sn: *Cicer arietinum* L.	plasma not specified (20 kV, 1 kHz, 0.5–30 min)	fungal decontamination	[36]
	surface micro-discharge (10 mW/cm^2^)	microbial reduction + surface of cotyledon + germination	[37]
	plasma not specified (10–50 s)	germination + fungal inactivation	[38]
**Common bean**, sn: *Phaseolus vulgaris* L.	DBD (8 kV, 510 W, 5–30 min)	fungi decontamination + germination + seed structure	[35]
	plasma not specified (20 kV, 1 kHz, 0.5–30 min)	decontamination of *Aspergillus* sp. and *Penicillium* sp. + seed germination	[36]
	micro-wave discharge (20 W, 15 s–20 min)	wetting properties + imbibition + germination	[47,48]
	capacitevly coupled plasma (13.56-MHz, 0–20 min)	coating of seeds + germination	[70]
**Egyptian clover**, sn: *Trifolium alexandrinum* L.	pulsed atmospheric-pressure plasma jet (10–20 kV)	morphological characters of two cultivars + fresh and dry yield	[84]
**Fenugreek**, sn: *Trigonella foenum-graecum* L.	plasma not specified (10–50 s)	germination + fungal reduction	[38]
	plasma jet (30 kV, 30 kHz, 10 s–15 min)	germination + early seedling growth	[60]
	DBD plasma jet (3.5–4 kV, 0–5 min)	elevated expression of diosgenin-related genes and stimulation of the defense system	[77]
**Grass pea**, sn: *Lathyrus sativus* L.	DBD (9.4 and 18.6 kV, 30 and 60 s)	food hydrocoloid aspects	[81]
**Honey clover**, sn: *Melilotus albus* Medik.	capacitively coupled rf discharge (5.28 MHz, 0.6 W/cm3, 5–20 min)	laboratory and field germination + decontamination	[41]
**Lentil**, sn: *Lens culinaris* Medik.	plasma not specified (20 kV, 1 kHz, 0.5–30 min)	decontamination of *Aspergillus* sp. and *Penicillium* sp. + seed germination	[36]
	micro-wave discharge (20 W, 15 s–20 min)	wetting properties + germination	[47]
	inductive RF plasma discharge (13.56 MHz, air, 18 W, 60 s)	seed surface + hydrophilization	[49]
	atmospheric pressure plasma jet (22.1 kV, 12 s)	PAW, germination + early growth + liquid fertilizer	[69]
	DBD (12 kV, 500 Hz, atmospheric ambient air)	PAW irrigation, early growth	[76]
**Mimosa**, 1st sn: *Mimosa caesalpiniafolia* Benth.	DBD (17.5 kV, 990 Hz, 3–15 min)	wettability + imbibition + germination	[50]
2nd sn: *Mimosa pudica* L.	Plasma jet (Ar, 10 kV)	movements of pinnules and petioles	[92]
**Mung bean** (syn. green gram), sn: *Vigna radiata* (L.) R. Wilczek	microwave discharge (2.45 GHz, 1.1 kW) produced PAW (5, 15 or 50 s), PAW sprouts treatment 0–5 min	bacteria inactivation on fresh sprouts by PAW	[45]
	gliding arc (air, 5 kV, 40 kHz)	PAW treatment, sprouts decontamination + antioxidant potential + total phenolic and flavonoid contents + sensory characteristics	[46]
	micro plasma jet (N_2_, He, air, and O_2_; 20 kV, 9 kHz, 25 W)	germination + early growth + catalase activity	[51]
	atmospheric pressure plasma jet (5 kV, 40 kHz, 750 W), 200 mL of PAW exposed 15–90 s	PAW or plasma treatments of one cultivar, germination + growth characteristics + total phenolic and flavonoid contents	[56]
	PAW atmospheric pressure plasma jet (air, O_2_, He, N_2;_ 30 mA, 30 min)	various PAW (O_2_, He, N_2_), germination + seedling growth + sterilization + physiological parameters + morphology SEM	[58]
	capacitively coupled RF discharge (13.56 MHz; 40 and 60 W)	germination + early growth + surface change + enzymatic activity	[61]
	DBD (O_2_, N_2_, air; 18 kV, 500 Hz),	treatment in PAW (direct, indirect), seed germination	[67]
**Pea**, sn: *Pisum sativum* L.	gliding arc (air, 15 kV)	drought resistance + seed germination + surface decontamination	[39]
	capacitively coupled rf discharge (5.28 MHz, 0.6 W/cm3, 5–20 min)	germination + early growth	[42]
	DBD (20 kV, 14 kHz, 400 W, 60–600 s)	surface + germination + early growth + physiology	[52]
	DBD (20 kV, 14 kHz, 400 W, 60–300 s)	early growth stages	[53]
	DBD (20 kV, 14 kHz, 400 W, 60–300 s)	DNA damage	[66]
	DBD (6 and 12 kV, 3 kHz, 1–10 min)	cultivar, physiology + germination	[62,86]
	capacitevly coupled plasma (13.56-MHz, 0–20 min)	germination	[70]
	DBD (20 kV, 14 kHz, 400 W, 60–300 s)	reduction of DNA damage	[82]
**Peanut**, sn: *Arachis hypogaea* L.	plasma not specified (air or SF6, 20 kV, 1 kHz, 0–20 min)	antifungal activity of seed surfaces + total aflatoxins	[40]
	plasma jet (4.4 kV, 70–90 kHz, 650 W, 3–5 min)	aflatoxin reduction	[43]
	rf plasma (13.56 MHz, 60–140 W, 15 s)	surface + germination + early growth + yield	[63]
	DBD (10–40 W, 1–15 min)	characteristics of antioxidant properties	[78]
	DBD (90 W, 0–5 min)	glycosylation conjugates of high-temperature peanut protein	[79]
	DBD (80 kV, 0–60 min)	antigenicity for defatted peanut flour and whole peanut	[88]
	DBD (90 W, 1–4 min)	peanut protein characteristics	[89]
	DBD (90 W, 1–10 min)	solubilization of peanut protein isolate	[90]
	DBD (not specified, 0.5–3 min)	glycation of peanut protein isolate and dextran	[91]
**Red clover**, sn: *Trifolium pratense* L.	capacitively coupled rf discharge (5.28 MHz, 0.6 W/cm3, 5–7 min)	germination + early growth + hormonal and flavonoid contents	[65,75]
**Soybean**, sn: *Glycine max* (L.) Merr.	plasma not specified (20 kV, 1 kHz, 0.5–30 min)	irrigation water on soybean sprout production.	[36]
	capacitively coupled rf discharge (5.28 MHz, 0.6 W/cm3, 5–20 min)	germination + early growth + surface decontamination	[42]
	PAW from DBD (not specified, air, 0–5 min)	PAW, sprout growth + aerobic microbe decontamination + phytohormone amount	[44]
	inductive RF plasma discharge (13.56 MHz, He, 60–120 W,15 s)	surface changing + seed germination + early growth + seedling physiology	[54]
	DBD (O2, N2, 25 kV, 50 Hz)	root growth + nodule formation + plant growth enhancement	[55]
	DBD (N2 or O2, 25 kV, 50 Hz)	physiological characteristics + vegetative growth + agronomic traits	[64]
	DBD (air, 80 kV, 50 Hz)	PAW, seed germination + plant growth + water uptake	[68]
	capacitevly coupled plasma (13.56-MHz, 0–20 min)	coating of seeds + seed germination	[70]
	DBD (20 kV, 14 kHz, 400 W, 60–300 s)	early growth stages + seedling physiology	[71]
	DBD (Ar, 10.8–22.1 kV, 3.4–15.6 W, 60 Hz, 12 s)	germination + early growth + physiology	[72]
	DBD (90 W, 0–27 min)	inactivation of soybean trypsin inhibitor in soymilk and Kunitz-type trypsin inhibitor in soybean	[80]
	pulsed power plasma discharge (50 Hz, 5 J/pulse, 27 kV, 10 A in pulse maximum, 20 s)	food quality (soybeans powder mixed with water)	[87]
	DBD (80 kV, 0–6 h)	treatment of soybean oil in a hydrogen gas environment at atmospheric pressure	[93]
	PAW gliding arc (air, 5 kV, 40 kHz, 750 W, 30–90 s)	PAW, microbial decontamination + food quality of bean curd (tofu)	[94]

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
