# Peer review of "Effects of Non-Thermal Plasma Treatment on Seed Germination and Early Growth of Leguminous Plants—A Review"

_plants, 2021, doi:10.3390/plants10081616_

Round 1
Reviewer 1 Report
General remarks
Dear Colleagues / Researchers!
The manuscript concerns mainly the effects of an innovative seed processing technology for agricultural purposes consisting in non-thermal plasma seed treatment (NTP). Contrary to standard seed treatment techniques, non-thermal plasma (NTP) treatments are effective (because they support optimal plant development), cheap, do not leave toxic residues and work mainly on the surface, thus avoiding damage to the embryo cells. Seeds are the first step of the agricultural cycle and at this stage, the plant can be given the highest probability of establishment, despite environmental conditions, to exploit the genetic potential of the seed. Furthermore, seedlings seem to be too sensitive to the oxidation of plasma and therefore, seeds seem to be the ideal target. In the light of this information, the authors of this review rightly focused mainly on discussing the immediate and long-term effects of plasma on the seeds of Fabaceae plants. The authors considered both commonly cultivated species of high economic importance and not-common plant species of regional importance. The manuscript consists of 9 chapters and, in my opinion, contains a valuable analysis of numerous literature data. Nevertheless, there are many inaccuracies in the manuscript text that need to be corrected or completed before it is published in Plants journal.The precise indication of the text fragments that raise my doubts is difficult due to the lack of numbered lines.
Acronyms/Abbreviations/Initialisms should be defined the first time they appear in each of three sections: the abstract; the main text; the first figure or table.
References: References must be numbered in order of appearance in the text (including table captions and figure legends).
Change the way of citing the literature in the entire mauscript, eg instead of "similar results and possible explanation presented [35] observed on…" write "similar results and possible explanation presented Runtzel et al. ... [35] "
Instead of "Finally, [44] studied the influence of PAW" write "Finally, Lee et al. [44] ... "
Detailed remarks:
Table 1 describes 18 plant species from the Fabaceae family, not the 20 as it appears in the title of the table.
In the table, instead of the purpose of the research presented in the cited publications, it would be better to include data on the results obtained by them (i.e. the observed seed plasma treatment effects). The research goal that the authors set for themselves is not always achieved.
The table shows examples not only of papers related to the treatment of seeds, but also of papers where the products obtained from them were the research material. So please modify the title of the table to suit its content.
Complete the sentence: Runtzel et al. [35] reported the effective fungal inactivation of Aspergillus parasiticus and Penicillium sp. on the surface of common bean after 10-30 min exposure. (exposure to what factor?)
Page 7 (line 6 from the top) - instead of Penicillinum sp. write Penicillium sp.
Page 7 (line 10 from the top) CFU stands for cell forming units, i.e. the number of microorganism cells capable of growing in the form of colonies on a microbiological medium. The term is obvious to microbiologists. Please delete the text in brackets: the unit is not explained in the original paper.
Page 8 (line 22 from the top) „Bußler et al. [59] studied the affection of pea seedlings (P. sativum ‘Salamanca’) flavonolglycoside profile, while considering the potential impact on their metabolic activity in different growth stages.” - Rewrite the sentence
Page 8 (line 37 from the top) instead of „increases concentrations of ……., superoxide dismutase, peroxidase, catalase” write „increase in the activity of the following enzymes superoxide dismutase, peroxidase, catalase”.
Page 8 (line 6 from the bottom)- I don't understand why the authors gave this information in parentheses?
Page 9 (line 3 from the top) “The content of carbonyl group, dityrosine cross-link and free sulfhydryl, secondary and tertiary structures, sodium dodecyl sul-phate–polyacrylamide gel electrophoresis, surface charge, surface hydrophobicity and solubility of PI were followed. The content ofcarbonyl groups, disulphide bonds, dityro-sine cross-link and surface charge in PI increased, which in turn led to a decrease in the absorption rate of protein into the water-oil interface. Part of the text copied from the abstract of the original work. Correct this excerpt specifying what the authors studied and what techniques they used, if applicable. Sodium dodecyl sulphate – polyacrylamide gel electrophoresis is a technique that is used to study certain characteristics of proteins or mixed peptides. What does the abbreviation PI stand for?
Page 9 (line 8 from the top) secondary structures - what compounds?
Page 9 (line 20 from the bottom) The authors state that "The NTP treatment has a positive effect on seed germination apparent as the acceleration of germination, the germination rate increasing and the seed dormancy breaking". On the other hand, examples of both positive and negative effects of plasma on the seed germination process are given below. Please eliminate this inconsistency.
Page 10 (line 26 from the top) „in [44], authors observed also the increase of ascorbate, asparagine, and γ-aminobutyric acid (GABA), partly also the cotyledon and hypocotyl, in soybean sprout during 1 to 4 days of farming” - complete and write this sentence differently
Page 11 (line 18 from the top) it is not the nodules that fix the nitrogen, but the bacteroids in them
Page 11 (line 20) In [54], „authors reported also the utilization of soybean seed reserve in improved by NTP”- rewrite the sentence.
In order to systematize the mechanisms that may be responsible for the observed effects of NTP used on legume seeds, it is worth reading the article: Waskow et al. Mechanisms of Plasma-Seed Treatments as a Potential Seed Processing Technology. Front. Phys., 14 April 2021 | https://doi.org/10.3389/fphy.2021.617345.
Author Response
Answers to the requirements of reviewer 1:
(all changes are marked in the author-coverletter-12824420.v1 file)
Acronyms/Abbreviations/Initialisms should be defined the first time they appear in each of three sections: the abstract; the main text; the first figure or table.
Thanks for the reminder. We have formally gone through the whole text and it is right in this respect now. Above all, we inserted explanatory terms in the headings of Figure 1 and Table 1 and checked the entire main text, but instead removed them from the Conclusion chapter.
References: References must be numbered in order of appearance in the text (including table captions and figure legends).
Thank you for this alert. We have resolved many inconsistencies by moving Table 1 to the end of the text before the Conclusion chapter. Thus, all references to the literature correspond numerically.
We checked the other links in the text. At first glance, it seems that the numbers jump meaninglessly in the text. This is because we describe one cited work in various chapters of the text, as it corresponds to the structure of the review. The numbering of the cited literature is fine now.
Change the way of citing the literature in the entire mauscript, eg instead of "similar results and possible explanation presented [35] observed on…" write "similar results and possible explanation presented Runtzel et al. ... [35] "
Instead of "Finally, [44] studied the influence of PAW" write "Finally, Lee et al. [44] ... "
Yes, this comment will significantly improve the comprehensibility of our text, thank you. Where appropriate, we made the proposed changes in the formulations.
Detailed remarks:
Table 1 describes 18 plant species from the Fabaceae family, not the 20 as it appears in the title of the table.
It is good that you have pointed out this discrepancy. There are 19 botanical species that we mention in the article. Everything is fixed now. In addition, we have better marked in the Table the place where there are two botanical species under one common name (mimosa). Now everything is clear.
In the table, instead of the purpose of the research presented in the cited publications, it would be better to include data on the results obtained by them (i.e. the observed seed plasma treatment effects). The research goal that the authors set for themselves is not always achieved.
Yes, I understand this comment. It seems like a good idea. However, our manual is designed so that it does not repeat the same data in different places. The chapters in the text deal with various issues (see Fig. 1) and their results, on the contrary, the table provides mainly an alphabetical list of monitored species (including scientific names and synonyms). In Table 1 we also present an overview of the used plasma apparatus and the object of the study for better orientation of the readers. The term Aim of Study was misleading, so we changed it to Object of the Study. This term better describes the content of the column in Table 1. The results of the performed experiments are described in the text.
The table shows examples not only of papers related to the treatment of seeds, but also of papers where the products obtained from them were the research material. So please modify the title of the table to suit its content.
Thank you for this very relevant comment. The new name is now: Table 1. Summary of studies performed on species of the Fabaceae family or on products thereof after non-thermal plasma (NTP) treatment.
We have also corrected the explanatory text under the title: The list is sorted alphabetically by plant species, 19 scientific species are mentioned. Abbreviations: syn. synonymous, sn scientific name, DBD dielectric barrier discharge, PAW plasma activated water, AC alternating current, RF radio frequency, USP universal stress protein, SEM scanning electron microscope.
Complete the sentence: Runtzel et al. [35] reported the effective fungal inactivation of Aspergillus parasiticus and Penicillium sp. on the surface of common bean after 10-30 min exposure. (exposure to what factor?)
Yes, thank you. The sentence was supplemented as follows: ..... surface of common bean after 10-30 min exposure of dielectric barrier discharge (DBD).
Page 7 (line 6 from the top) - instead of Penicillinum sp. write Penicillium sp.
Thank you very much for your care. The term has been rewritten.
Page 7 (line 10 from the top) CFU stands for cell forming units, i.e. the number of microorganism cells capable of growing in the form of colonies on a microbiological medium. The term is obvious to microbiologists. Please delete the text in brackets: the unit is not explained in the original paper.
The abbreviation CFU appears in the text in 3 other places. Readers of Plants journal are representatives of a wide representation of various scientific disciplines. Therefore, we leave the explanation of the colony-forming unit (CFU) in the text.
Page 8 (line 22 from the top) „Bußler et al. [59] studied the affection of pea seedlings (P. sativum ‘Salamanca’) flavonolglycoside profile, while considering the potential impact on their metabolic activity in different growth stages.” - Rewrite the sentence
Yes, thank you. The sentence has been changed as follows: Bußler et al. [59] studied the affection of pea seedlings (P. sativum ‘Salamanca’) flavonol glycoside profile after DBD treatment, while considering the impact on their metabolic activity in different growth stages.
Page 8 (line 37 from the top) instead of „increases concentrations of ……., superoxide dismutase, peroxidase, catalase” write „increase in the activity of the following enzymes superoxide dismutase, peroxidase, catalase”.
Yes, thank you. The sentence has been changed.
Page 8 (line 6 from the bottom)- I don't understand why the authors gave this information in parentheses?
We replaced the abbreviation STI with "soybean trypsin inhibitor" both in Chapter 4 (in the new version Chapter 6), but also in Chapter 8.
Page 9 (line 3 from the top) “The content of carbonyl group, dityrosine cross-link and free sulfhydryl, secondary and tertiary structures, sodium dodecyl sul-phate–polyacrylamide gel electrophoresis, surface charge, surface hydrophobicity and solubility of PI were followed. The content ofcarbonyl groups, disulphide bonds, dityro-sine cross-link and surface charge in PI increased, which in turn led to a decrease in the absorption rate of protein into the water-oil interface. Part of the text copied from the abstract of the original work. Correct this excerpt specifying what the authors studied and what techniques they used, if applicable. Sodium dodecyl sulphate – polyacrylamide gel electrophoresis is a technique that is used to study certain characteristics of proteins or mixed peptides. What does the abbreviation PI stand for? Page 9 (line 8 from the top) secondary structures - what compounds?
We have significantly simplified the whole part, which concerns the work of the authors Mehr and Koocheki 2020. Abbreviation PI has been removed from the text. New version:
Mehr and Koocheki [81] investigated structure and emulsifying properties of grass pea (Lathyrus sativus) protein isolate after DBD treatment. The content of carbonyl group, dityrosine cross-link and free sulfhydryl, secondary and tertiary structures, sodium dodecyl sulphate–polyacrylamide gel electrophoresis, surface charge, surface hydrophobicity and solubility of grass pea protein isolate were followed. Overall, the results indicated that cold plasma treatment had positive effect on the interfacial and emulsifying properties of grass pea protein isolate in terms of thermodynamic stability of protein on interface, globulin dissociation, and increase in oil-droplet surface electrical charge.
Page 9 (line 20 from the bottom) The authors state that "The NTP treatment has a positive effect on seed germination apparent as the acceleration of germination, the germination rate increasing and the seed dormancy breaking". On the other hand, examples of both positive and negative effects of plasma on the seed germination process are given below. Please eliminate this inconsistency.
Correct comment, thank you for the warning. Our strict statement has been corrected.
Page 10 (line 26 from the top) „in [44], authors observed also the increase of ascorbate, asparagine, and γ-aminobutyric acid (GABA), partly also the cotyledon and hypocotyl, in soybean sprout during 1 to 4 days of farming” - complete and write this sentence differently
Yes, we changed the sentence as follows: „in [44], the authors also observed an increase in ascorbate, asparagine and γ-aminobutyric acid (GABA), and followed the development of cotyledon and hypocotyl in germinating soybean seed of soybean from the 1st to the 4th day of cultivation.”
Page 11 (line 18 from the top) it is not the nodules that fix the nitrogen, but the bacteroids in them
Yes, correct observation, we changed the sentence as follows:....., indicating that mutualistic bacteria in the nodes fixed nitrogen more actively than the control.
Page 11 (line 20) In [54], „authors reported also the utilization of soybean seed reserve in improved by NTP”- rewrite the sentence.
Yes, we changed the sentence as follows: In [54], authors report the root weight increased by 27 % in seed soybean after NTP treatment and the soluble sugar and protein contents were 16 % and 25 % higher than those of the control.

Reviewer 2 Report
In the present work, the authors investigate the application of non-thermal plasma treatment (NTP) of seeds on seed dormancy releasing, germination induction and early growth seedlings of Legume spp.
This research is interesting and clearly provides new data valuable for the research community. The manuscript has high formal standard.
GENERAL COMMENTS:
The paper title is well stated, it is informative and concise.
ABSTRACT, INTRODUCTION, AND CONCLUSIONS
Abstract and introduction is well written. In spite of that I have a few objections against its present form:
-there were no chapters short characterizing of seed dormancy and germination- the seeds are the starting point for determining the effect of NTP on seed dormancy and germination, and seedling growth ;
-the order of the chapters: first seeds, then seedlings;
-sequence in the sentence - we cannot talk about germination until dormancy has been broken;
-please systematize: non-thermal or non thermal.
In Coclusion chapter please refer to all aspects of NTP application as presented in the manuscript.
The items of literature included in the paper are adequate to the subject of the paper.
Author Response
Answers to the requirements of reviewer 2:
(all changes are marked in the author-coverletter-13122544.v1 file)
I have a few objections against its present form: -there were no chapters short characterizing of seed dormancy and germination - the seeds are the starting point for determining the effect of NTP on seed dormancy and germination, and seedling growth ; -the order of the chapters: first seeds, then seedlings;-sequence in the sentence - we cannot talk about germination until dormancy has been broken;
The chapter dealing with influencing seed germination in Leguminum by non-thermal plasma is called 4. Seed Germination and contains a large amount of information. We've moved the Seed Germination chapter from 5th to 4th position to better fit the logical order of the text. This order is definitely better, thank you for the warning.
This chapter also includes the issue of dormancy breakage using non-thermal plasma, specific information relates to the work on Mimosa caesalpiniafolia (da Silva et al. 2017).
-please systematize: non-thermal or non thermal.
Good comment, thank you, we fixed on non-thermal throughout. We left the original form non thermal, nonthermal or non-thermal in the citations of the original articles.
In Coclusion chapter please refer to all aspects of NTP application as presented in the manuscript.
We have added the relevant aspects as set out in the text and in Figure 1. The new version of the Conclusion is now clearer. This comment improved the final impression of our manuscript.

Round 2
Reviewer 2 Report
This paper by Šerá et al. has clearly benefited from the revision, as advised by reviewers.
Author Response
Thank you very much for your comments.